# Effects of the COVID-19 Pandemic on Sustainable Food Systems: Lessons Learned for Public Policies? The Case of Poland

**Michał Dudek**  **and Ruta Śpiewak** * 

Institute of Rural and Agricultural Development, Polish Academy of Sciences, Nowy Świat 72 Street,
00-330 Warsaw, Poland; mdudek@irwirpan.waw.pl
* Correspondence: ruta.spiewak@irwirpan.waw.pl

**Abstract:** COVID-19 has affected the functioning of food systems all over the world. This paper seeks to identify and analyse the economic, legal and institutional, as well as social effects of the pandemic's outbreak on food systems, and the implications for the EU Farm to Fork Strategy whose main purpose is to put food systems on a sustainable path. Qualitative economic and social impact analysis was used to identify the above types of effect on the food system on a macroscale, using Poland as an example. Information was sourced from existing data and qualitative studies. Studies show that the consequences of the pandemic for individual elements of the food system in Poland in 2020 were related to numerous disruptions in functioning, leading to uncertainty, financial losses, and interrupted transactions. The crisis under analysis also revealed modifications in these actors' behaviours in food markets, noticeable in changes in consumption patterns and in the ways demand for food was met. Nevertheless, an analysis of the gathered information and data testifies to the food system's relative resistance to the effects of the pandemic, and also to the adaptive skills of the system's entities, especially food producers and consumers. The paper's discussion contains recommendations for public policies shaping the food system, pointing to actions that might reduce the negative effects of other potential exogenic crises in the future and aid the implementation of the Farm to Fork Strategy's principles.

**Keywords:** sustainable food system; COVID-19 pandemic; farmers demand and consumption patterns; EU Farm to Fork Strategy

## 1. Introduction

The COVID-19 pandemic has changed the functioning of food systems all over the world. Its outbreak caused a shake-up in the operations of all of the food system's actors, starting from the sector providing the means of food production, through food producers, the processing industry and logistics, all the way to consumers [1]. Research conducted so far indicates that the severity of the pandemic's effects varied depending on the region of the world, the level of food market development, the wealth of societies, the type of link in the system, and the response from public institutions influencing the system's character [2]. As an external crisis, the pandemic affected the food system directly and indirectly. For instance, research [3] showed how COVID-19-related disruptions in transportation (e.g., access to container transport) and demands for related services (such as retail food pickup and delivery services) impacted food supply chains. The coronavirus crisis had also an impact on the health of the system's participants and on their behaviours [4].

Therefore, the state of danger caused by the disease triggered a response from governments, which introduced legal and institutional measures aimed at limiting the spread of the disease or supporting persons and businesses particularly strongly affected by the new epidemic situation [5,6]. Such actions from public policies have deeply transformed food system functioning. One of the examples of such intervention was stricter occupational

health and safety procedures at the workplace (OHS), that affected the functioning the agri-food sector as well [4].

The literature of the subject includes various assessments of the magnitude and longevity of the effects of COVID-19 on food systems. According to some researchers, the impact of the crisis has reduced environmental pressures in the short term, mainly leading to a slowdown in economic growth in labour-intensive sectors such as agriculture and services [7]. Others underline that the effects of the coronavirus have had such a strong impact on food systems that, in the future, these will not function the way they used to [8,9], and that this is an opportunity to create transformative public policies serving to build more sustainable food systems, and also enabling the food system innovations emerging during the pandemic to be maintained and developed [10]. The present paper seeks to capture direct and indirect manifestations of the coronavirus pandemic's impact on the food system in an economic, legal and social aspect, on the example of Poland during the first 12 months of the pandemic. The authors also trace the responses of public policy to the crisis from the perspective of the Polish food system, particularly its supply and demand aspect. Their observations have led them to offer recommendations for public policies involving the food system, and will also enable a better response to be given to exogenic shocks of this kind in the future. The conclusions from the material gathered will also afford suggestions for the development of instruments serving the implementation of the EU's Farm to Fork Strategy.

The paper aims to identify and analyse the economic, legal and institutional as well as social effects of the COVID-19 pandemic on food systems, and the implications for the implementation of the EU Farm to Fork Strategy, especially for the Central and Eastern European Countries (CEECs) countries as they struggle with quite similar challenges regarding the food system.

The information used in the study was sourced from the literature of the subject and from publicly available data and information on changes in the economic circumstances, conditions of operation and attitudes of the food system's actors, especially agricultural producers and consumers, that emerged in the first year of the pandemic.

## 2. COVID-19 Implications for the Farm to Fork Strategy

Although agriculture is responsible for approximately one-fourth of greenhouse gas emissions in the EU, the measures taken so far in the field of agricultural policy in the EU have not brought about the desired changes from the point of view of mitigating the climate disaster [11]. The Farm to Fork Strategy, which is part of the European Green Deal (EGD) being implemented by the EU, is aimed at creating a sustainable food system that will guarantee food security while also ensuring access to a healthy diet produced in a way that is safe for the planet. It is the first such policy to combine various food-related strategies, in which the consumer becomes the policy's pivotal element [12]. The Farm to Fork Strategy assumes that all the food chain participants, from raw material producer to end consumer, have to play a role in the construction and operation of a sustainable food system [13]. Among other things, the strategy emphasises sustainable animal production, reduced use of pesticides and mineral fertilisers, shortened food chains, and the increased importance of expanding the knowledge of all the food chain participants, with a special focus on farmers. There have also been many critical voices indicating that provision of the EGD is not sufficient to introduce the necessary changes [14,15].

The authors of a paper published in The Lancet have appealed for the development of a "social vaccine" in addition to the biological one—tackling economic, environmental and social determinants of health—to overcome the current global crisis [16]. The EGD currently being developed, and with it also the Farm to Fork Strategy, provides an opportunity to strengthen the European organism in areas such as climate issues, food safety, social integration, etc. The conclusions to be drawn from the effects of the COVID-19 pandemic, indicating a strong correlation between environmental problems, health and the economy, should find a place in the implementation of the EGD.

COVID-19 has shown many weaknesses of food systems based on long, industrial, specialised chains with strong dependency on foreign workers [17]. At the same time, the crisis caused by the pandemic has created opportunities for redesigning the food system to be more sustainable and resilient [18].

Researchers from all over the world list three main issues in relation to agri-food production and the supply chain in the context of the COVID-19 pandemic, which correspond to the assumptions of the Farm to Fork Strategy:

- People today follow a healthy diet to a greater extent than in the past, in order to protect themselves and their immune system, hence the increased demand for functional food containing bioactive ingredients [19,20];
- More attention is now given to food safety, to prevent the spread of the coronavirus among producers, people involved in food processing, retailers, and consumers [21];
- Fears about and actual difficulties with ensuring food security for some population groups have appeared in the face of lockdowns and restrictions on movement [22,23].

The above remarks, which—as will be shown further on—remain relevant for both Polish and global food chains, justify the necessity to implement the changes contained in the Farm to Fork Strategy despite voices suggesting that this will cause a substantial decrease in food production in the EU, increasing food production costs as well as increased imports of agri-food products from third countries.

## 3. Materials and Methods

A variety of data on the food system's functioning in Poland in 2020 were used to achieve the study's objective. They were mainly related to empirical materials describing the economic situation of individual entities within the food system, especially agricultural producers (conclusions from reports, economic situation data), legal measures directly and indirectly affecting food production and consumption (laws, directives, communiqués), and also information published in reports and on the internet, presenting the behaviours of the food system's individual actors (themed web portals, social media).

The method used to analyse the gathered empirical material was a qualitative, expert economic and social impact analysis conducted at the macroeconomic and social level [24]. In the most general terms, this method involves an analysis (estimate) and assessment of the socio-economic effects, both positive and negative, caused by specific changes or interventions, most often implemented as part of a policy or by a government. The authors decided on a triangulation of research methods, as a verification process that improves the accuracy of analyses by taking several viewpoints into consideration [25]. This is particularly important when a new situation is being explored, one that has been poorly identified or not at all; the pandemic with its scale and its impact on food chains definitely fits this description. The next section of the paper introduces a wider context for our analysis by giving an overall description of Polish food system to better understand different COVID-19 impacts that are considered later on.

## 4. Results

*4.1. Agri-Food Sector in Poland—A General Picture*

In order to present an analysis of various consequences of COVID-19 pandemic for the Polish food system a brief presentation of the domestic socio-economic context is essential. It should be highlighted that an agri-food sector is of high economic, social and environmental significance. In 2019, the share of primary sector (agriculture, forestry and fisheries included) in Gross Value Added (GVA) in Poland is about a half higher than in the EU-28 and amounted to 2.6% (Table 1). At the same time, according to the official statistics a considerable proportion of economically active people worked in agriculture in the country. Employment in this sector as a proportion in total employment was high and stood at the level of 9.1% (Table 1). This is one of the reasons behind a low level of labour productivity in agriculture, which despite an upward trend in last years, is significantly under an average EU level. In 2019, an average agricultural factor income per full-time

worker in Poland amounted to EUR 6,9 thousand and was by 158% lower than in the EU (EUR 17.9 thousand/AWU) [26].

When describing Polish food system, it is important to point out that the majority of farms in the country (about 99%) is family farms. This group is highly polarised, diverse in production specialisation and includes about 1.4 million agricultural households, with an average size of 11.3 ha of utilised agricultural area [27]. Over two-thirds of them are non-market-oriented units with very small economic potential. Their users usually do not make a living from agricultural activity, but rather from paid work or social benefits received from the insurance system [28]. On the other hand, about one-fifth of agricultural holdings (about 250–290 thousand farms) are economically strong units, utilising almost two-thirds of the country's arable land and capital resources, as well as producing four-sixths of domestic agricultural output [29]. In recent years in Polish food production, the simplification and specialisation processes of those farms have taken place.

**Table 1.** Selected characteristics of the Polish agri-food sector, 2012–2019.

| Specification | 2012 | 2013 | 2014 | 2015 | 2016 | 2017 | 2018 | 2019 |
|---|---|---|---|---|---|---|---|---|
| Share of primary sector in GVA (in %) | 3.9 | 3.2 | 2.9 | 2.5 | 2.7 | 3.2 | 2.4 | 2.6 |
| Employment in agriculture (% in total employment) | 12.2 | 11.8 | 11.5 | 11.2 | 10.2 | 9.6 | 9.5 | 9.1 |
| Value agri-food export (billion EUR) | 17.9 | 20.4 | 21.9 | 23.9 | 24.3 | 27.8 | 29.7 | 31.8 |
| Balance in agri-food trade (billion EUR) | 4.3 | 6.1 | 6.7 | 7.8 | 7 | 8.5 | 9.7 | 10.5 |
| Number of enterprises in food industry | 15,726 | 14,218 | 14,625 | 16,028 | 15,899 | 16,831 | 16,912 | 17,640 |
| Labour productivity (GVA in thousand Euro, in current prices/employed person) | 25.3 | 27.2 | 27.7 | 29.6 | 30.1 | 30.8 | 31.1 | n.d. |

Source: own elaboration based on [26,27].

The group of bigger, commercial farms, apart from family labour input, hired non-family, seasonal labour force (mainly farms specialised in fruit and vegetable production, estimated to employ over 300 thousand seasonal migrant workers from non-EU countries [30]), as well as leased the agricultural land from other land owners. This category of agricultural holdings, together with small in number but highly productive agricultural enterprises (e.g. agricultural cooperatives, producer groups and companies), are strongly linked with the market and, collaborating intensively with other actors of the agri-food chain (enterprises selling agricultural inputs, food processing plants and manufactures, food services and retail) [31]. As a whole, Polish agri-food system has long been going through dynamic structural changes, which were initiated by economic transformation and continued over the course of European integration by inclusion under the CAP mechanisms [31].

Over the last three decades food sector in Poland was under process of dynamic modernisation and concentration thanks to foreign direct investments (FDI) as well as financial support from the EU founding [32]. In 2019, within this segment of the economy operated over 16,9 thousand enterprises, mainly micro and small units. As a result, the Polish food sector became an important, efficient, export-oriented branch of the national economy with a significant contribution to the national global output (in 2018, the sector's share in GDP amounted to 3.2%) and to the employment (336 thousand persons employed in 2019) (Table 1). From 2004 to 2019, Polish export of food products grew six times, from EUR 5.2 to 31.5 billion [32]. In 2019, the positive balance of trade in agri-food products stood at EUR 10.5 billion [32]. When it comes to food system distribution, Poland is in the process of increasing concentration of companies toward the emergence of large retail units, with the number of small stores decreasing by about 5% per year [33]. The major contribution in total food sales had 295 big companies (international corporations included) [26].

Organic farming might be an important element in the implementation of sustainable development principles within food systems because it delivers not only private goods

(such as high-quality organic food), but also environmental public goods, such as landscape, biodiversity and quality of natural resources [34]. According to EGD, by 2030, 25% of arable land in the EU should be organic. In Poland, we can observe in last years a decline of organic farming at a rate unseen in EU. Recently organic land consists of less than 3% of all arable land [35]. At the same time, organic market has grown, with import being a key contributor. The low level of social capital of farmers is another weak point of Polish agriculture, which is manifested by the fact that only 15% of farmers belong to agricultural cooperatives [36], and farmers themselves are also reluctant to cooperate with each other at an informal level [37].

### 4.2. Economic Effects of the COVID-19 Pandemic for the Polish Food System

The economic consequences of the pandemic had a varied impact on different elements of the economic system. In the case of the food system, whose global value is estimated at 3 trillion US dollars (approx. 10% of the global GDP), the influence of the SARS-CoV-2 pandemic was symmetrical and asynchronous, because it affected the markets' demand and supply sides simultaneously, while its acuteness was felt by various groups of entities at different points in time [38]. COVID-19 affected the functioning of all elements of the system and the relationships between them. It thus exerted all kinds of pressure and caused disruptions in a complex organism involving the following segments:

- Supply of means of agricultural and food production;
- Primary production of agricultural raw materials and products;
- Agri-food processing;
- Wholesale and retail food trade;
- Marketing, logistics and transport;
- Food preparation and consumption;
- By-product management;
- Edible energy.

The pandemic also changed the situation in markets linked to food management, which include the market for means of production and the labour market, causing their imbalance, mainly on the supply side. This was due to the fact that agricultural production is a sector sensitive to deliveries of varied resources. At the same time, the demand for food during the pandemic was affected by the overall macroeconomic situation defined by fluctuations in financial and currency markets as well as growing pessimism in the consumer mood [39]. Furthermore, many countries limited their international trade, including agri-food trade (in part taking advantage of the pandemic situation to ensure increased sales for their domestic output). This was accompanied by the emergence of significant demand pressure, leading to insufficient food deliveries in various places around the world, which translated into growing prices for certain products in some regions (meat, fruit and vegetables) [40].

Generally speaking, at the agricultural producer level and as regards the labour market, the pandemic and resultant restrictions on business operations were seen to affect more labour-intensive agricultural segments (e.g., fruit and vegetable production) more acutely, while having a relatively weaker impact on capital-intensive segments (e.g., dairy, poultry or field-crop production). Nevertheless, problems with deliveries for ongoing production, i.e., fertilisers, feed, fuel and pesticides, were reported in both cases. This situation was caused by reduced exports of farming chemicals from the main producers (China) to countries with industrialised agricultural production [41]. Limited availability of capital assets used in agricultural production (machines and equipment) was also observed. Alongside growing costs of business operations during the pandemic, problems that also emerged included work organisation issues at farms (a labour deficit), problems with the sale of goods, maintaining financial liquidity, and completing investment projects [42].

The restrictions on the population's social activity introduced by most countries (restrictions on movement and contacts) transformed the food demand segment, in particular changing the way food was bought and consumed [1]. Generally speaking, COVID-19 con-

tributed to the emergence of non-traditional forms of obtaining and eating food, strengthening the online sales channel and retail outlets, including places for trading in local products, while weakening the HoReCa sector (hotels, restaurants, bars, catering businesses) [43]. At the same time, increasing social isolation during the pandemic led to changes in the way consumers spent their time, strong fluctuations in demand for foodstuffs ("panic buying"), and an increased proportion of spending on food in household budgets, caused by people's drive to stock up and secure their future needs [10].

The shock on global food markets caused by SARS-CoV-2 was superimposed on the sector's overall favourable situation created by a sufficient level of supplies, decreasing fuel prices and positive agroclimatic forecasts. Irrespective of this, the economic crisis caused by the pandemic had a negative effect on the demand side, in both the short and the long term, by worsening the income situation of food consumers due to reduced wages and increased unemployment. The deteriorating financial condition and decreasing food consumption as a result of the crisis caused by COVID-19 is estimated to have affected poor households to a greater extent. Another consequence of the situation was a changed demand structure in food markets (the population's changed diet) in favour of relatively more easily available products with a longer shelf life (easier to store and transport), such as highly processed foods and grain products, coupled with lower demand for meat [25].

The pandemic had a negative impact not only on the global economy, but on Poland's economy, too. The force of this negative effect was substantial, though varied depending on the sector or industry. Poland as a country with a relatively high share of the agri-food sector in the GDP and in foreign trade, which is characterised by a diverse product range, underwent temporary shake-ups in trade exchange as a result of limitations affecting imports of agricultural means of production and raw materials, limited inflow of seasonal workers, and reduced sales of agri-food products [43]. Similarly to the whole Polish economy, the situation in food production deteriorated slightly in the first quarter of 2020, which was typical for the winter season. This situation was due to a drop in farms' cash incomes as a result of smaller sales volumes and to a worsening confidence among agricultural producers [43]. This deterioration was relatively more noticeable for farms specialising in the production of non-perennial crops [43].

However, the sector's situation worsened dramatically in the second quarter of 2020 (Figure 1). The underlying causes included reduced demand for agricultural raw materials and foodstuffs (mainly from the HoReCa sector, which was the most severely affected by the restrictions), and also the decreasing dynamics of wages in relation to retail food prices. The serious deceleration of the dynamics of international agri-food trade due to the restricted flow of goods between different countries also had a significant impact, and translated into an unfavourable situation in most segments, since they had been strongly integrated with external markets [39,43]. The prices of agricultural products decreased as a consequence. The restrictions connected with the growing incidence of SARS-CoV-2 infections were felt by a significant group of agricultural producers. Depending on the study, they were seen to have a negative impact on between 44% and 75% of farmers, and among them mainly on the group specialising in animal and mixed production, reducing their income and thus leading to reduced spending on current assets for production, on machines and equipment, and on investments in buildings and structures [44]. A deep slump was noticeable in all of Poland's regions irrespective of the type of farming. The relatively greatest slowdown was reported on farms growing perennial plants (economic situation index down by 15.6 points). The situation was not improved by the seasonal rise in fruit, vegetable and potato prices. The fulfilment of orders for fruit deliveries (especially apples) was impeded by a shortage of workers, many of whom had returned to Ukraine for fear of the pandemic [45].

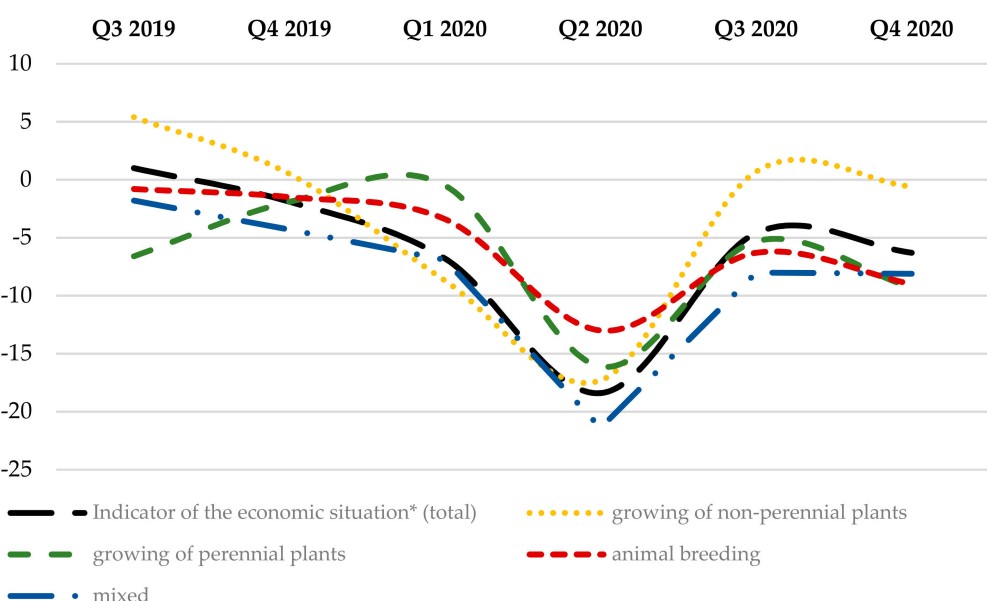

**Figure 1.** Economic effects of the COVID-19 pandemic on Polish food producers in 2020. * The indicator reflects the economic situation in agriculture and in selected types of farming. Source: own compilation based on data [43].

The third quarter of 2020 brought a partial recovery that made up for the losses of the previous period. The economic situation in food production was influenced not only by the lifting of restrictions, but also favourable weather and the transferred effects of trade turnover revival from February and March [43]. Relatively speaking, the improvement of the situation mainly applied to farms specialising in non-perennial crops (economic situation index up by 18 points). Agricultural raw material and product prices were seen to increase, especially those for fruit and vegetables. The market disruptions were partially evened out by growing demand for food on the domestic market, less restrictive access of foreign nationals to the Polish labour market, and intensifying exports.

The last three months of 2020 brought another deterioration of the economic situation for agricultural producers, mainly as a result of the worsening public confidence, which was closely linked to the second wave of the pandemic and a return to harsher restrictions in socio-economic life (Table 2). Regardless of smaller demand for food from the HoReCa sector, the producers of these goods, like other farmers, reported relatively high cash incomes as a result of favourable purchase prices, which were increasing thanks to growing domestic consumer demand and international demand (Figure 1) [43].

Despite the negative economic effects caused by the pandemic in 2020 for food producers, the financial situation of households in Poland slightly improved. Although households gained higher incomes than in 2019, they limited the scale of their spending due to the crisis. In real terms, households' expenditures on consumer goods and services were lower by 6.2% [44]. However, the expenditures on food and non-alcoholic drinks have increased. Their share in total expenditures increased by 2.6 p.p., i.e., from 25.1 to 27.7% [44]. At the same time, during the first year of the pandemic in Poland, there was a change in the frequency, volume and structure of purchases made by households. For instance, there was an increase in the consumption of most food staples, relatively most flour (by 18.6%), butter (by 14.1%) and cheese and cottage cheese (by 6.6%). Expenses on food services decreased by 26.7%, which resulted from the closure of restaurants [44].

**Table 2.** Dynamics of new COVID-19 cases and the relevant policy responses in Poland.

| Specification | Q1 2020 | Q2 2020 | Q3 2020 | Q4 2020 |
|---|---|---|---|---|
| Average number of new COVID-19 cases | 83 | 353 | 621 | 1308 |
| Stringency Index * | 16.6 | 72.4 | 37.7 | 63.4 |

* Average value of the Stringency Index, i.e., a composite measure based on nine policy indicators which record information on countries' containment and closure policies, rescaled to a value from 0 to 100 (100 = strictest) [46].

*4.3. Legal and Institutional Effects of the COVID-19 Pandemic for the Polish Food System*

The COVID-19 pandemic completely changed the legal and institutional conditions in which businesses, employees and consumers functioned. In the face of the risk of infection to their citizens, most countries in the world, including Poland, decided to take extraordinary action, among other things freezing the economy (lockdowns) as an element of public crisis management [47]. The essence of this type of action, which was motivated by the exceptional situation (a state of higher necessity), lay in granting public authorities additional prerogatives while also restricting the scope of and rights related to individual freedom, e.g., the right to move around, ownership rights, and the right to pursue business operations. These measures came in the form of ordinary legal means and included regulation by acts of law and the relevant directives issued on their basis [47–49]. Such normative acts became the legal basis for fighting the epidemic (a directive of the government minister for health introduced particularly drastic restrictions of freedom and constitutional rights by regulating issues reserved for acts of law).

In connection with the first wave of the pandemic, a state of epidemic danger was introduced in Poland as of 12 March 2020 (changed into a state of epidemic a few days later), which was connected with border traffic checks. Border checks on the internal borders of the Schengen Area were lifted as of 13 June 2020, including sanitary checks and mandatory quarantine. However, restrictions on entering Poland and mandatory quarantine were upheld for citizens of third countries. Harsher restrictions on socio-economic life started being reintroduced in August (the COVID-19 "second wave"), but they only covered individual powiat/county units depending on the intensity of new cases. Counties were divided into two zones, yellow and red, each with a different range of restrictions. As the number of new cases increased all over the country, on 24 October 2020 the whole of Poland was covered by the restrictions applicable to red zones (Table 2). As a result of continued high numbers of COVID-19 new cases and deaths, restrictions in social and economic life continued until the end of 2020.

The aim of special legislative measures introduced in Poland starting from March 2020, their restrictiveness changing depending on the scale of infections and deaths, was to limit the spread of the disease and protect public health (Table 2). Frequent and hasty amendments made to laws and directives aimed at implementing measures to fight the pandemic only increased the institutional and social chaos, and suggested a lack of public administration bodies' coherent concept for alleviating the symptoms of the crisis [50,51]. The norms in force in connection with the COVID-19 pandemic might be divided into universally binding regulations and recommendations, and guidelines targeted at specific entities in the socio-economic system. In both cases, they mostly involved various dos, don'ts and restrictions [52]. The new and frequently amended regulations affected the modes, forms and types of operation of all entities within the food system in Poland, starting with suppliers of means of agricultural production, through agricultural producers, transport and logistics, buyers of agricultural raw materials and products, to trade businesses, as well as influencing food consumer behaviours.

The functioning of all the actors of the food chain was significantly affected by regulations that were aimed at fighting and containing the COVID-19 pandemic and restricted trans-border traffic between Poland and other countries. Restrictions on cross-border traffic involving non-EU member states, especially Ukraine, had particularly unfavourable effects because they reduced the mobility of employees. Poland's borders were closed. Those who had been allowed to cross the border included citizens of other countries who had a Polish

work permit. However, when a state of epidemic was introduced in Poland (March 2020) and an extraordinary situation was announced in Ukraine, the visa-issuing procedure for Ukrainian citizens wishing to enter Poland was suspended. It was resumed from the end of April [53]. The pandemic also caused problems with the international flow of goods important for the agri-food sector's functioning. One measure that made things much easier for suppliers of such goods within the EU was the "green corridors" enabling border procedures within the Trans-European Transport Network (TEN-T) to be simplified and speeded up [54].

From the point of view of the food supply chain's functioning, one important factor was the pandemic-related extension of the sanitary service's prerogatives. In a situation of danger to public health, sanitary inspectorates could issue recommendations, guidelines and decisions obligating various entities to undertake specific preventive or control measures, actions related to the distribution of certain products, and to cooperate with other public administration bodies.

In connection with danger to public health, on 16 April 2020 the Chief Sanitary Inspectorate (GIS) issued guidelines for farmers and plantation operators on measures to prevent the spread of SARS-CoV-2 [55]. Unlike the social distancing obligation introduced in the pandemic's early stages, the ban on movement did not apply, among others, to anyone moving around in order to pursue agricultural operations or perform jobs on a farm, nor to anyone performing tasks aimed at protecting and securing crops and farm produce. The otherwise required wearing of face and nose coverings did not apply to farmers during work on the farm and when remaining on private land around their home. On the other hand, if there were people from outside the family working on the farm, social distancing and protective coverings were required. The sanitary service's guidelines also included a ban on working on farms for anyone who was sick or showed symptoms of sickness as well as the obligation to maintain proper hand hygiene, apply hand disinfectants frequently, maintain proper airway hygiene, wash and disinfect work surfaces where food was produced, observe one's own health and act appropriately if symptoms appeared.

Additionally, in connection with the pandemic, on 8 and 25 May 2020, the sanitary service and the Ministry of Rural Development and Agriculture simultaneously issued guidelines for agricultural producers who employed foreigners [56]. The new procedures aimed to guarantee the safety of seasonal workers and farmers, enable foreign workers to complete a 14-day quarantine, guarantee their safe work and stay in Poland, and also ensure stable production on farms.

The conditions under which sites for selling agri-food products, such as marketplaces and bazaars, could function were very important for the operations of agricultural producers and for food deliveries directly to consumers. The sanitary service issued the relevant guidelines for marketplace and bazaar operators on 25 March 2020 [57]. These mainly required site managers as well as food producers and vendors to follow rigorous hygiene rules.

Dealing with the outbreak of the COVID-19 pandemic, the public authorities obligated businesses with employees, including entities from the food sector, to ensure appropriate distances between work stations and provide personal protection relevant for fighting the epidemic [58]. Special guidelines recommended by WHO for food sector businesses were announced as well [59]. These guidelines were aimed at preventing the spread of COVID-19 in food industry workplaces, among other things by the use of gloves, physical distancing, following specific procedures if infection was established, and following practices recommended for shops and the transport sector. Other actions undertaken by the sanitary service included the publication of information and warnings important for the functioning of food deliveries in Poland (e.g., encouraging health-promoting behaviours, warning against foodstuffs allegedly protecting people from SARS-CoV-2 infection and contributing to curing COVID-19; information on the European Food Safety Authority's stance suggesting a lack of evidence that food might be a source or indirect link in the chain of virus transmission).

The outbreak of the COVID-19 pandemic and the closure of the economy caused restrictions in the work of institutions and public administration bodies responsible for the functioning of organisations important for the operations of agri-food businesses, including agricultural producers. Early on in the pandemic, such institutions suspended direct visits from clients, recommending remote contacts, e.g., by telephone or email. When the number of new cases stabilised (May/June), direct visits were resumed, but under a strict sanitary regime.

The response to the pandemic in Poland was not limited to the public authorities' measures aiming to prevent, counteract and fight the spread of the virus. A number of interventions and changes were also undertaken to support the economy and its individual sectors. The predicted unfavourable effects of the pandemic on the agri-food economy and food supply chains, in the form of a wave of bankruptcies and growing unemployment, induced the public authorities to intervene. Many countries around the world responded to the pandemic by launching special aid for this sector. The most important support mechanism for the whole economy in Poland, including the agri-food sector, was a project called the Anti-Crisis Shield. Under this mechanism, measures adopted exclusively for entities operating in the food supply chain were varied and included social insurance premium exemptions, employment subsidies, controlled wholesale and retail prices and margins, tax breaks, deductions of taxpayers' losses incurred as a result of the pandemic in 2020 from their income for 2019—on condition that their income had decreased by at least 50%, support in employing workers (exemption from paying their social insurance premiums), the extension of foreigners' residence and work permits for the duration of the epidemic, easier access to e-administration services, postponed loan instalment payments, preferential credit and loan terms, measures increasing work time flexibility for entrepreneurs operating in the agri-food sector.

*4.4. Social Effects of the COVID-19 Pandemic for the Polish Food System*

In the social aspect, the COVID-19 pandemic is observed to have had a number of negative effects on the food supply chain, the most widespread and acute being a shortage of labour and reduced food security. Health problems coupled with restrictions on the movement of employees caused at least a temporary shortage of labour in every component of the food chain. Research shows that this led to especially serious disruptions in animal breeding, horticulture, and grain processing, i.e., segments that are among the most labour-intensive [60]. The shut-down of production in various areas of the economy resulted in the loss of jobs by a large group of the population, with an estimated 660,000 people losing their jobs in Poland in the first three months of the pandemic [61]. Consequently, the number of people who lost their financial liquidity also grew, pushing them to the brink of poverty and jeopardising their chances of satisfying even basic needs related to proper nutrition during the pandemic. Moreover, access to food also worsened due to the restrictions on movement, potentially leading to a serious decrease of food security for some social groups, i.e., the elderly, the less wealthy, those living in smaller, poorly equipped and more isolated localities. In addition, during the first lockdown, Polish people listed the following as being the most difficult for them: the necessity to stay home (48%), not being allowed to enter forests and parks (45%), and the necessity to wear face masks (44%). As less important limitations, respondents mentioned the closure of most shops (25%) and the closure of clubs and restaurants (22%) [62]. The unusual situation also caused a change in the attitudes and behaviours of consumers and food producers.

One might distinguish several key consumer attitudes that emerged in the pandemic: (1) the group fearful of the pandemic and the virus itself went on impulsive, big shopping sprees in the early stages of the lockdown, to get a grip on their fears; (2) forced to limit eating out, some consumers started cooking at home more often; (3) sales of plants increased, including herbs and vegetable seedlings, which could be explained by a growing interest in growing food at home (controlled, safe, readily available) as well as people's need to arrange a green space around themselves (giving them a sense of being close to nature) [20].

People with higher incomes showed an increased interest in growing their own food and buying larger amounts of fruit and vegetables. Since the authorities recommended a minimal number of visits to retail outlets (especially over long distances), the number of people making their purchases at local shops increased [20]. In the longer term, these behaviours could turn out to be just temporary, as a direct effect of the pandemic situation. Some changes, however, e.g., increased online purchases, could change consumers' food supply patterns permanently. Those among the population who had not previously been involved in virtual shopping, but now were forced to consider it by the lockdown and fear of infection, might have developed new skills in the present situation and come to see the advantages of using this alternative form of buying what they need.

In the first half of 2020, over 3000 new online shops emerged in Poland. The country is among the leaders in online shopping. The Chamber of the Electronic Economy's research shows that 14% of internet users buy food online [63]. According to internet users, the food sector coped best with the pandemic situation and the restrictions it involved. A similar trend is observed among farmers, given that the group of producers offering and selling food online is growing steadily. It is still too soon to speak of the full implementation of technical or social innovations responding to the crisis caused by the pandemic. Nevertheless, some interesting trends have been observed, revealing the directions in which Poland's food sector will develop. A comparative study of three countries shows that— unlike Italy—the United States and Poland saw an increased interest in local food during the lockdown [64]. According to research carried out in connection with the present report, farmers who had not used online sales tools before did not attempt to go in this direction by themselves, but new entities appeared or existing ones developed to support farmers in reaching consumers directly. One reason why farmers showed little online activity was that there can still be problems with internet access in Poland—84% of rural households had internet access in 2019—and with the quality and capacity of internet services [65].

One way that agricultural producers and food consumers could be connected in Poland during the pandemic was via the "Polish e-Bazaar" web portal (www.polskiebazarek.pl, accessed on 2 August 2021) [66]. Another new project connecting big-city consumers and producers was KARMNIK, an internet platform delivering foodstuffs from a very specific region, i.e., Podlasie and eastern Mazovia, while also promoting its own local product brand (Figure 2). This was an interesting example of activity serving to shorten the supply chain and supporting the (territorially strictly defined) production of quality local food. KARMNIK's founders say that the spark to undertake this kind of project was COVID-19, which they consider to be a time when people started getting more interested in food, "entered the kitchen" and at the same time were unable to eat out. One of the project's initiators had lost her job elsewhere, which became an additional incentive to move into new areas.

Producers themselves pursued varied and innovative activities with the aim of reaching consumers directly. One option was to run a Facebook fan page showing what was going on at the farm, in detail and on a day-to-day basis. This is a way to build producer–consumer relationships based on trust and full reliability, even during a pandemic. One example of such activity was a farmer breeding free-range chickens: via Facebook, she posted regular updates on what was happening at her henhouse and what she was feeding her chickens (Figure 2). This gave consumers the feeling that they knew exactly what kind of product they were buying and could contact the producer directly. An idea currently being tested at a cooperative called Spółdzielnia Ostoja is innovative on a large scale. It is a system of "vegemats", i.e., parcel lockers meeting the requirements for storing organic vegetables [67]. This is a good solution for a time like the pandemic, when people have to limit social contacts, because product ordering and delivery is contact free (Figure 2).

The founder of the web portal My Zbieramy [68], which hooks up farmers and fruit growers with customers who are willing to pick their purchases themselves and pay less than at the greengrocer's or the supermarket, saw it as a way of resolving the growing problem of a labour shortage. This project enabled less food to be wasted, brought savings

for the consumer pickers, and enabled farmers to gather at least part of their crop. The project had an educational aspect as well. In addition, farmers proposed overnight stays or promoted attractions on their farms in their ads. "Both parties see actual people in each other, and not just boxes with anonymous goods", the portal's founder has remarked [69].

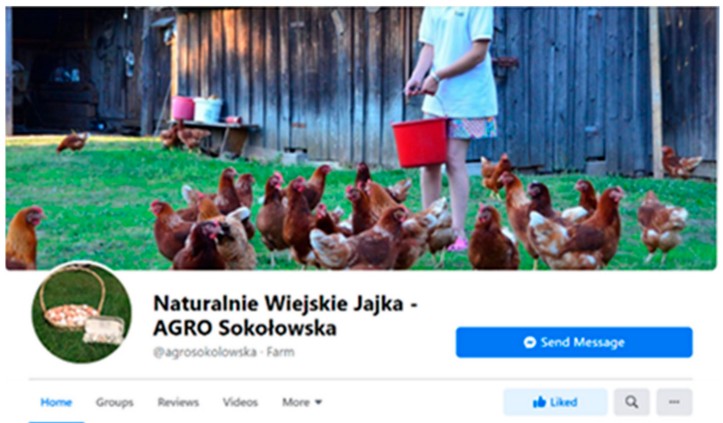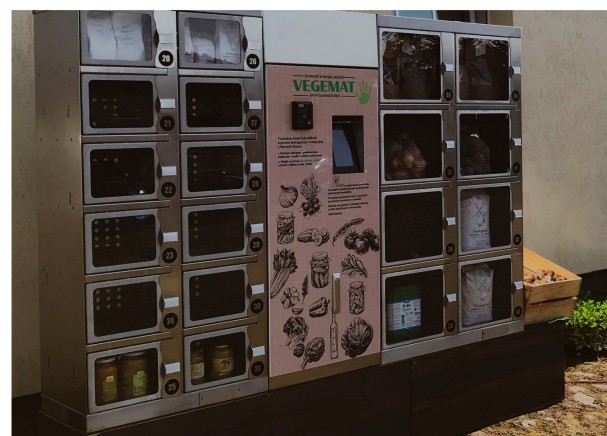

**Figure 2.** Examples of social initiatives within the food supply chain during the COVID-19 pandemic in Poland (KARMNIK on the left and Spółdzielnia Ostoja on the right). The source of picture on the left is: https://www.facebook.com/agrosokolowska (accessed 12 December 2020) and the source for the picture on the right: https://ostojanatury.pl/bio-hub/ (accessed 29 December 2021).

Analysing the social effects of the COVID-19 pandemic, it is necessary to draw attention to the problems that some consumer groups (social groups) have with obtaining food, i.e., to maintaining their food security. The number of food insecure people increased globally during the pandemic [70]. This was also the case in Poland: one study [71] shows that the number of people declaring they are sometimes short of money to buy food grew during the pandemic. However, we also know that both big and small food producers joined in projects supporting those most affected by the pandemic, i.e., the elderly as well as healthcare workers. For example, the Wzywamy Posiłki initiative distributed 220,000 meals among hospital employees in the first six months of the pandemic. Food support was provided to over 10,000 senior citizens in the same period.

## 5. Discussion

The COVID-19 pandemic changed the functioning of the entire food system in Poland as well as the whole world [71,72]. The closure of the economy and the resultant economic slump, restrictions on international trade and halted demand for goods were factors with a negative impact on the financial foundations of domestic agricultural producers, the processing industry, transport, energy and trade businesses [73]. The long-observed deficit of workers, especially in seasonal primary production, was exacerbated in the new situation. Serious challenges included adjusting to various sanitary norms, which were introduced and frequently changed by the public authorities, and to work and production organisation guidelines, but also upholding and rebuilding existing and establishing new business relationships. Another difficult test was finding the right response to strongly fluctuating domestic demand and to the changing shopping behaviours of food consumers. The present analysis of information related to the functioning of the agri-food sector during the COVID-19 pandemic in Poland has shown the sector to have been relatively resistant to the crisis.

From the perspective of food chains, the experiences gathered during the pandemic should be an important point of reference for the coming years in the implementation of the EGD, and especially the Farm to Fork Strategy. The pandemic has shown how important shortening the food chains is for the food system's sustainability (in every aspect) and

resilience. As the research cited here shows, in a crisis situation, short food chains turned out to be better and more resilient from the point of view of consumers and farmers alike.

The sense of danger connected with the new virus increased demand for quality food, which also includes food produced with less pesticides. The growing need for this kind of food is compatible with another important objective of the strategy under discussion, i.e., limiting the use of pesticides and artificial fertilisers.

Achieving most of the goals of the Farm to Fork Strategy, including increasing the acreage dedicated to organic farming, requires a substantial workforce. The results of our analyses show that even today, the availability of agricultural workers is a serious problem that will only increase over time and with successive stages of the European Green Deal's implementation. It is therefore necessary to support the employment of seasonal workers from outside the EU.

As the data cited in the present paper show, the financial results (income situation) and mood of domestic agricultural producers worsened in the second quarter of 2020, i.e., the time of the first lockdown and the introduction of government restrictions on the economy. The sector saw an improved economic situation and made up for part of its losses in the next quarter. This was the result of completed transactions that had been suspended due to the pandemic, cleared international trade channels and the inflow of workers from neighbouring countries, an easing of previously imposed economic restrictions and, to the least extent, the public authorities' launch of indirect and direct support for agricultural producers.

According to the data and information analysed here, the COVID-19 pandemic brought mostly negative as well as, much less often, neutral effects for operations conducted by individual entities within the food system. First of all, the coronavirus situation significantly increased people's uncertainty, not only in relation to their personal situation and fears for their own health and that of their families and employees, but also in relation to their jobs and making a living. Secondly, the most painful effect of the pandemic for food producers was the reduced availability of workers, including the services of seasonal workers. A great many entities in the food chain were dependent on workers from Ukraine, whose shortage was especially acute between March and May 2020. Seriously limited access to labour translated into financial losses stemming from reduced or destroyed agri-food crops. The problems declared by businesses from the food supply chain also included decreasing demand for their products and problems with liquidity caused by delayed payments from buyers. Low level of cooperation between farmers has been another factor hindering the operation of food systems during the pandemic crisis.

It is worth noting that the negative consequences of the pandemic did not affect all the elements of the system in the same way. Relatively more strongly negative effects of the pandemic and its restrictions involved those entities whose increased need for workers fell on spring 2020 or whose labour needs were relatively constant. At the same time, regardless of the scale of agricultural production, a relatively more favourable financial and market situation in the pandemic was noted when entities used diverse sales channels for their products, and especially if they were involved in direct sales of food to consumers and in processing their farms' own produce.

Available research and an analysis of the literature help indicate effective ways for the food system's actors to deal with the changing socio-economic situation (best practice) during the COVID-19 pandemic, including various institutional social and economic barriers/restrictions emerging as its consequence. They include the following:

- Establishing, maintaining, developing and shortening direct relations with food consumer customers;
- Increasing the added value of offered products by increasing their health benefits, specifying production locations (local product), highlighting the products' flavour value, producing food in an environmentally friendly way (certified organic food);
- Diversifying the channels and ways of selling products (middlemen, processing businesses, wholesalers and retailers, own sales outlets, including online shops);



- Pursuing a pro-employee hiring policy (e.g., appropriate wages, good work conditions, maintaining long-term relationships with employees);
- Launching food deliveries directly to customers (food boxes);
- Following sanitary norms related to the pandemic during food production, such as social distancing, disinfection, wearing mouth and nose coverings, quarantine, extended work hours and shift work as well as increasing the emphasis on compliance with food safety and work safety rules.

The cited best practice examples have contributed to food system entities in Poland building resistance to crises similar to the COVID-19 pandemic. At the same time, they suggest there is a need to develop a varied range of activities, through both top-down actions (by national-level institutions) and grassroots activity at the producer and local food system levels, with the aim of increasing their stability and adaptability. Such new activities might include:

- Social innovations related to work style and forms of employment, but also projects supporting the most disadvantaged social groups (e.g., the elderly, the poor);
- Technological innovations improving the standard of hygiene at food supply, production, distribution and consumption sites;
- e-technology innovations enabling full producer confidence to be built in a way allowing customers to gain an insight into the entire chain of production;
- e-commerce innovations involving the use of new technologies, not only at the stage of selling to the end customer, but also at the middleman and producer stages;
- Innovations in e-communication between public offices/institutions and farmers, e.g., introducing digital reporting.

The data analysis outlined in the present paper has enabled the authors to draw up some preliminary recommendations for public institutions involved in the functioning of Poland's food system, including the government, ministries, public bodies, local governments, and agricultural consultancy centres. The focus is on developing relatively more optimal institutional conditions for the operations of agricultural producers during a socio-economic crisis like the COVID-19 pandemic:

- Providing support in finding workers for food sector businesses, both domestically and internationally, including improving agricultural employees' work and living conditions; one major form of aid for the sector is for the government to cover the costs of coronavirus testing among agricultural producers' workers; promoting vaccinations, developing clear and simple procedures and instructions for foreign workers on the possibilities and terms of working on Polish farms;
- Gathering, organising and efficiently distributing information on the legal conditions of running farms and other entities in the food supply chain during times of sudden and powerful social, climate and market changes, e.g., a pandemic; as research shows, the actors of the food supply chain were disoriented by various restrictions related to the spread of the virus, which were frequently changed (e.g., the rules and possibilities of inviting and employing seasonal workers from other countries, rules of quarantine, testing of workers to check for coronavirus infection);
- Limiting bureaucracy related to requirements for agri-food production, including expanding the possibilities for online contacts with public institutions and digital gathering of required documentation;
- Disseminating information on the conditions, possibilities and ways of conducting direct sales of agri-food products (including processed foods) and the potential economic benefits of such sales methods;
- Disseminating economic knowledge related to new economic models of running businesses, focused on selling food directly to consumers, among other options, and highlighting the vision and goals of such operations, using the internet, digital technologies and new product delivery methods;

- Disseminating knowledge on soft skills in management and marketing related to building deep and long-term relationships with customers and with employees;
- Supporting the formation of direct sales sites such as marketplaces as well as, for example, parcel locker systems adapted to the requirements of food sales, supporting the development of e-commerce innovations related to food trade and processing;
- Continuously improving agricultural consultants' competence in supporting farmers in the production of quality food, including organic food, and innovative methods of direct sales, and also in developing means and strategies of dealing with crisis situations; ensuring active consultants involved in rural development and focused in their work on teaching soft, market and marketing skills as well as skills involving advanced internet technologies connecting producers and customers;
- Gathering examples of best practice in the production of quality food and in shortening food chains, and publicising them on the web;
- Facilitating and promoting the cooperative and/or producer group model of operation; organising training in soft skills, with the aim of overcoming any reluctance to undertake joint operations (teaching cooperation by showing examples and best practice);
- Educational campaigns to build awareness of the health and environmental benefits of organic food among consumers, including the youngest public.

## 6. Conclusions

The period of turbulence caused by the effects of the COVID-19 pandemic has shown that short food supply chains are advantageous to consumers and producers alike, for economic as well as organisational reasons. The question of how to effectively implement the idea of bringing consumers closer to producers, which is desirable in the Farm to Fork Strategy, and how to improve food quality and safety and sustainability of the food system, requires further in-depth studies and an evaluation of earlier findings. At the same time, it will be a serious challenge to make sure that, as food quality improves and environmental pressures on food production diminish, people in the EU and beyond are guaranteed food security. The fact is, the crisis caused by the pandemic has led to an increase in the number of food insecure people in Poland and most other places around the globe.

Creating a climate neutral EU economy based on elements like sustainable agriculture and food chains and edible energy, as assumed in the EDG, requires the development of a number of instruments at the national as well as the EU level to ensure stable farmer incomes coupled with a significant decrease in the environmental costs of food production, processing and distribution. The way food systems functioned in Poland during the pandemic revealed the weakness of institutions from the broadly understood environment in which agriculture operates, which might make it difficult to build and implement such regulations.

It has to be acknowledged that there are several limitations of our paper. First, we analysed situations that had been changing dynamically, in order to repeal the further dynamics of change. Hence, it would be worth turning again to food producers (farmers) and sellers with the question of how they perform after two seasons of COVID restrictions. We also present the analysis conducted for only one country—Poland. However, CEECs are facing similar challenges within their food systems, and the recommendations provided in our paper might be especially useful for different food systems' stakeholders from this region.

**Author Contributions:** Conceptualisation: M.D.; methodology: M.D. and R.Ś.; formal analysis and investigation: M.D. and R.Ś.; writing—original draft preparation: M.D. and R.Ś.; writing—review and editing: M.D. and R.Ś.; funding acquisition: M.D. and R.Ś.; resources: M.D. and R.Ś.; supervision: M.D. and R.Ś. All authors have read and agreed to the published version of the manuscript.

**Funding:** The study was supported with European Union funds under Scheme II of Technical Assistance "National Rural Network" (KSOW) of the Rural Development Programme for 2014–2020. The Managing Authority of the Rural Development Programme for 2014–2020 is the Minister of

Agriculture and Rural Development. The European Agricultural Fund for Rural Development: Europe Investing in Rural Areas. This study was also supported by Polish National Science Centre, grant number UMO-2017/26/D/HS6/0083.

**Institutional Review Board Statement:** Not applicable.

**Informed Consent Statement:** Not applicable.

**Data Availability Statement:** Part of the data supporting the reported results can be found at https://ssl-kolegia.sgh.waw.pl/pl/KAE/struktura/IRG/publikacje/Strony/Koniunktura-w-rolnictwie.aspx (accessed on 10 August 2020).

**Acknowledgments:** The paper makes use of part of the results of research carried out by Fundacja Badań Wiejsko-Miejskich RURall (Rural and Urban Research) and commissioned by the Agricultural Advisory Centre in Brwinów (CDR) in the project titled: "Developing a Research Tool and a Preliminary Analysis of the Process of Shortening the Food Supply Chain in the Context of the COVID-19 Pandemic", implemented as part of an operation of the Innovation Network in Agriculture and Rural Areas (SIR) called From Farm to Fork: Analysis of the Process. The authors wish to thank Adam Czarnecki, PhD hab., and Monika Stanny, PhD hab., for their input during work on the project.

**Conflicts of Interest:** The authors declare no conflict of interest.

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
