# Peer review of "Effects of the COVID-19 Pandemic on Sustainable Food Systems: Lessons Learned for Public Policies? The Case of Poland"

_agriculture, doi:10.3390/agriculture12010061_

Round 1

Reviewer 1 Report

The manuscript proposes an interesting study on the effects of the COVID-19 pandemic on the food system from both the supply and demand perspective.

The manuscript is well organized and fits well with the journal’s topics. However, it also presents several criticalities that need to be addressed.

Firstly, the text is quite cumbersome and needs to be revised to make it easier to read.

In the introduction, the overview of the impact of the COVID-19 pandemic on the agriculture sector should be augmented. Actually, unlike other types of activities, it has almost never been interrupted because it is an essential sector, as stressed in https://doi.org/10.3390/safety7030052. Hence, the pandemic has caused numerous modifications both at the operative level (e.g. working procedures) as well as along with the food supply chain network (e.g. in https://doi.org/10.1111/cjag.12235). The discussion of these aspects can allow a clearer context analysis of the study.

Also section 2 should be augmented to better clarify the “farm-to-fork” strategy and its relevance in the EU context. In particular, the sentence in lines 90-92 and the following list should be expanded, providing more details and information on the mentioned sources.   

In section 4, additional figures/tables showing the data related to the analysis carried out could help the reader in better understanding the Polish context.

In the discussion of results a critical analysis of the study limitations has to be included (e.g. the fact that the analysis is related to one country only). Moreover, practical implications and lessons learned from the Polish experience should be discussed more in details.

The Authors should follow the editing rules of the journal for the figures’ captions and the references in the text.

Author Response

Dear Reviewer,

we highly appreciate your valuable contribution and would like to thank you for your feedback and remarks. We discussed every single issue that has been pointed out in your review. Your comments and our reactions are listed below in the following table.

For more details, please see the revised word-file version and the attached list of responses to your comments. We also supplemented the text with changes suggested by the another reviewer. We hope that the introduced corrections will be sufficient to publish our manuscript.

Best regards,

The authors

Reviewer 2 Report

The paper examined the economic, legal and social effects of Covid 19 in Polish food system focusing on agriculture.  The topic of the paper is actual, and relevant, the content of the article fits well to the scope of this scientific journal.

The paper is interesting, but at the present form descriptive, and statements are very general, some parts are not understandable for readers not familiar with Polish economy, mainly role and structure of agriculture and food industry. Just some examples for missing information:  sectors:    differences between labor intensity, seasonality, production cycle , farm size, ownership, share of family farms, the employee structure local and foreigners, share of Polish people go to abroad for agricultural seasonal worker, market channels, export and import dependence both on final products  and inputs. In my mind, these information and the showing the changes due to the Covid 19 can make some general statements more justified. The question also is the effect differences on regional level.

On the economic side some information also would be useful, how the income has changed in general and how the spending on food. The price effects   also important on producer and consumer side as well.

The sensitivity of health issues also very important, but we do not have information about the Polish consumers’ behavior towards organic food for examples.  Therefore, we could understand the effect and responses knowing the baseline in details.

My suggestion is that the authors should pointed out weaknesses and vulnerability of Polish food system and agriculture, how the pandemic highlighted these problems and highlighted the importance of EU Farm to Fork Strategy.

The authors should be justified in more details.

Author Response

(The authors gave the same response as above.)

Round 2

Reviewer 1 Report

The Authors have satisfactorily improved the manuscript. Hence it can be considered for publication.

Reviewer 2 Report

All of my suggestions were taken into consideration,  in the present form of the paper   represents not just high scientific level but understandable for international readers  as well. Thanks for authors and congratulate! It can be considered for publication in present form.